# Spermiogenesis, Stages of Seminiferous Epithelium and Variations in Seminiferous Tubules during Active States of Spermatogenesis in Yangzhou Goose Ganders

**DOI:** 10.3390/ani10040570

**Published:** 2020-03-28

**Authors:** Muhammad Faheem Akhtar, Ejaz Ahmad, Sheeraz Mustafa, Zhe Chen, Zhendan Shi, Fangxiong Shi

**Affiliations:** 1College of Animal Science and Technology, Nanjing Agricultural University, Nanjing 210095, China; faheem_dear@hotmail.com (M.F.A.);; 2Department of Clinical Sciences, Faculty of Veterinary Sciences, Bahauddin Zakariya University, Multan 60800, Pakistan; ejazvetrohail@gmail.com; 3Key Laboratory of Control Technology and Standard for Agro-product Safety and Quality, MOA, Institute of Animal Science, Jiangsu Academy of Agricultural Sciences, Nanjing 210014, China; chenzzju@163.com

**Keywords:** stages of spermiogenesis, cellular associations, Yangzhou goose ganders

## Abstract

**Simple Summary:**

Optimum hatchability and fertility in the goose industry is inevitable without emphasizing on reproductive health of ganders. Determining stages of spermatogenesis is a critical factor in evaluating the age of maturity and reproductive health of animals. The current study was designed to evaluate the steps of spermiogenesis and stages along with morphological changes in seminiferous epithelium in Yangzhou goose ganders. The results of the study revealed that inside seminiferous epithelium, initial steps of spermiogenesis are depicted by changes in acrosomic granules, whereas further stages are identified by nuclear morphological changes. Furthermore, ten steps of spermiogenesis, nine stages of seminiferous epithelium and four types of spermatogonia Ad, Ap1, Ap2 and B were identified. In Yangzhou goose ganders, recommended age of sexual maturity is 227 days of age.

**Abstract:**

The past three decades revolutionized the goose industry in the world. China holds the world’s largest goose breeds stock by 95% of the global total. To optimize the goose industry and cope with ever increasing poultry meat and egg demands, there is a dire need to focus on reproduction, as most geese breeds exhibit poor reproductive performance. The present study was conducted with the aim to add a contribution in the goose industry and research by the histological visualizing step wise development of germ cells during spermatogenesis by microscopy and a histological technique. Yangzhou goose is a synthetic breed developed by using local goose germplasm resources of China. It is popular in the Chinese goose industry due to high productivity and adaptability. This research evaluated the steps of spermiogenesis and stages along with morphological changes in the seminiferous epithelium in Yangzhou goose ganders. For the assessment of various stages of the seminiferous epithelium cycle, testis sections were embedded in molten paraffin wax. The initial steps of spermiogenesis were depicted by changes in acrosomic granules, whereas further stages were identified by nuclear morphological changes. Ten steps of spermiogenesis and nine stages of seminiferous epithelium were identified. Four types of spermatogonia Ad, Ap1, Ap2 and B were recognized. The results depicted a clear variation in the diameter of seminiferous tubules (ST), epithelium height (EH), luminal tubular diameter (LD), number of seminiferous tubules per field and the Johnsen score. Microscopy indicated that the stages of seminiferous epithelium were similar to other birds and mammals and the ST diameter, EH, LD and Johnsen score are positively correlated while the number of seminiferous tubules per field is negatively correlated with the ST diameter, EH, LD and Johnsen score. Fertility in Yangzhou ganders can further be improved by visualizing the histological development of germs cells in testis tissues during spermatogenesis after onset of breeding season and maturity. Our results suggest that Yangzhou ganders reach complete sexual maturity at 227 days of age.

## 1. Introduction

The goose industry in China is especially prominent with an annual output accounting for approximately 95% of the world’s total production: with over 600 million geese slaughtered and more than 250 million tones of meat produced [1]. Due to increased market demand for day old goslings in the goose industry and the seasonality of the goose production, the improvement of reproductive performance in geese is crucial [2]. To optimize the reproductive performance and for high economic returns, balance between two major key reproductive factors, i.e., hatchability and fertility, is inevitable. Genetics, breeder age, mating activity, nutrition, housing system, male to female ratio and breeding seasonality are common sub factors of hatchability and fertility. From an economic point of view in commercial poultry production, efficiency of the male breeding stock (cocks, ganders and toms) is measured in terms of the sperm quality factor, which includes the ejaculate volume, the sperm concentration and the percentage of live spermatozoa [3,4]. All the goose breeds used in China are seasonal breeders, which may differ in breeding seasonality depending on their native habitat or locations [1]. In commercial practice, domestic breeder geese are used for hatching egg production for up to the fourth breeding season [5]. Thus it is suggested to use experienced ganders with one year old geese to achieve a higher fertility in the first laying cycle [5,6]. In song birds, testosterone concentrations in the breeding season (late spring) elevated 3.5 folds than the non breeding season (early spring) [7]. In Magang ganders, testosterone concentration reached peak levels in the onset of the breeding season and subsequently declined [8]. Testosterone concentrations remain higher in the breeding season, which is important to nourish the development of germ cells (spermatogonia, spermatocytes and spermatids) in seminiferous tubules. The efficiency of spermatogenesis is the estimated number of spermatozoa produced per day per gram of testicular parenchyma [9]. Spermatogenesis is a complex developmental process by which diploid spermatogonia generate haploid spermatozoa through a series of cyclic and highly coordinated events known as spermatocytogenesis, meiosis, spermiogenesis and spermiation [10]. During spermatocytogenesis, a spermatogonium (Ad germ cell) undergoes mitosis to generate new Ad and Ap1 cells, which further divides into two daughter Ap2 cells. Each Ap2 cell divides into two B spermatogonia, which themselves each divide into two primary spermatocytes [11]. Spermiogenesis is the transformation of spermatids into sperm cells without further cell division. Spermiation is the release of the sperm from Sertoli cells into the seminiferous tubule lumen [11]. The cycle duration is not similar among species or even in different strains of the same species. For example the cycle duration is 12 days in the Sherman rat [12], 9 days in the Sprague-Dawley rat [13] and 10 days in the bandicoot rat [14]. Similarly, 12–13 days in the chicken [15], 13 days in the Japanese quail [16], 14 days in the guinea fowl [17] and 12 days in the turkey [18,19]. 

The main morphological features of testes in three different phases of development and/or activity are sexually matured testis, sexually matured and active testis and sexually matured but resting testis [11]. Histomorphological alteration in seminiferous tubules is dependent on these maturation stages. In most mammals, cross-sections of seminiferous tubules contain varying generations of germ cells arranged in a definite, successive, repeatable order, from the basement membrane to the lumen of the tubule [20,21]. Each generation of germ cells is at exactly the same step of development and normally associated with other generations in a manner that is predictable and constant for the species [22]. In spermatogenesis, each step is fundamental, since defects that occur in any one of them can result in the failure of the entire process [23]. This has been studied extensively in mammals and most of our knowledge regarding developmental and transformational processes in the spermatid is derived from these studies [24].

In birds, the cellular association in the seminiferous epithelium occupies only a small area of the seminiferous tubule. Therefore, it is difficult to determine the composition of the associations and consequently the cycle of the epithelium and spatial arrangement of stages of the cycle in the seminiferous tubule [25]. Overall, the various stages of spermatogenesis in avian species appear to be of shorter duration than the corresponding stages in mammals [26]. For example, while the time from the onset of meiosis to the end of spermiogenesis is about 26 days in the mouse [21], 29.5 days in the ram [22], 37 days in the bull [26] and 45.5 days in human [27], it is only 14 days in the fowl or drake [19,20], 11 days in the quail [25] and 14 days in the guinea-fowl [26]. The Yangzhou goose (*Anser domesticus*) is the synthetic breed developed by using the local goose germplasm resources of China. This breed is widely used in China due to its high production traits for fertility, hatchability, egg production and meat production [28]. This breed reaches the maturity weight at 160–170 days of age but the age at sexual maturity is unknown. The histological evaluation of germ cells development during spermatogenesis in Yangzhou ganders could help us increase our knowledge in this field. Therefore, the objective of this study was to analyze the evolution in spermatogenesis in the seminiferous epithelium of Yangzhou goose ganders at 180 day, 200 day and 227 day of age. By depicting various stages and observing morphological changes in germ cells development, we can evaluate (I) the efficiency of spermatogenesis, (II) the reproductive stage, i.e., mature or immature testis and (III) the histopathology of the testis tissue, which may guide avian histopathologists, researchers and masses affiliated to the goose industry to improve overall reproductive efficiency and fertility of ganders, specifically.

## 2. Materials and Methods

### 2.1. Ethics Statement

The housing of Yangzhou ganders and experimental protocols were conducted in accordance with the Guide for the Care and Use of Laboratory Animals prepared by the Institutional Animal Care and Use Committee of Nanjing Agricultural University, China (Approval Numbers: 31572403 and 31402075). 

### 2.2. Experimental Location

The experiments were carried out at Sunlake Swan Farm (119°58′ E, 31°48′ N), Henglin Township, Changzhou, Jiangsu Province, China. Changzhou lies 31.76463° N and 119.9409° E. Temperature varies from 0 to 32 °C in winter and summer respectively. Rainfall is monomial with annual rainfall ranging by 1000–1100 mm and peaks from June to July.

### 2.3. Birds and Management

A flock 30 Yangzhou ganders (*Anser anser*) at 161 days of age, having same genetic origin, were used for experiments. The experiment was conducted in April-June 2016. Birds were individually identified using tags placed through their inner wings to prevent detection by other birds and to avoid pecking. Birds were maintained at ambient temperatures between 25 and 32 °C until the end of the experiment. The ganders had free access to drinking water. They were fed ad libitum with a mixed feed of 12.5% crude protein, supplemented with green grass whenever possible. Feed was offered during daytimes. The lighting schedule was 11 h light and 13 h dark until the end of the experiment. 

### 2.4. Testes Sample Collection

In Yangzhou ganders, testis maturation is different from body maturation. That is why days 181, 200 and 227 of age were selected for testes samples collection. For testes collection, ten birds were slaughtered at 181 days of age. On day 200, ten birds were slaughtered for a second time and on the 227th day of age, the remaining ten birds were slaughtered. Birds were sacrificed by cervical dislocation thrice during the course of the experiment. Immediately after slaughtering and testes samples collection, testes tissues were frozen in liquid nitrogen, and stored at −80 °C.

### 2.5. Experimental Protocol and Microscopy

A sliced piece of testicular tissue (0.125 cm^3^) from the left testis of each gander was taken and immediately fixed in 10% buffered neutral formalin solution for 24 h and used for histological evaluation by using an automated tissue processor (LEICA RM 2235). The fixed tissues were dehydrated in alcohol of ascending concentrations, i.e., 70%, 80%, 90% and 100%, and absolute alcohol, respectively, cleared in xylene (two changes), unfiltered and embedded in molten paraffin wax. Tissue sections (5 μm) were cut perpendicular to the longest axis of the testis, mounted on glass slides and stained with hematoxylin and eosin (Nanjing Jiancheng Bioengineering Institute, Nanjing, China). Stained sections were individually examined under a bright field Olympus BX63 light microscope (OLYMPUSBX63, Olympus Corporation, Tokyo, Japan) at 10× and 40× magnification for changes in the diameter of the seminiferous tubule, numbers of spermatogonia, spermatocytes and elongated spermatids.

### 2.6. Germ Cells Identification and Stereological Analysis

Spermatogonia were mainly distinguished based on nuclear characteristics, heterochromatin appearance and distribution [29,30]. In all slides, germ cells were identified on the basis of the relative size, shape and nuclear morphology. Identification of the various stages of the seminiferous epithelium was facilitated by microscopically analyzing morphological changes in the germ cells’ developmental stages and cellular associations. 

### 2.7. Diameters of the Seminiferous Tubule (ST) and Tubular Lumen (LD), Epithelium Height (EH) and Number of Seminiferous Tubules Per Field (No. ST/Field)

For the measurement of histological parameters, tissues sections (5 µm) were cut perpendicular to the longest axis of the left testis, mounted on glass slides and stained with hematoxylin and eosin (Nanjing Jiancheng Bioengineering Institute, Nanjing, China). Stained sections were individually examined under a bright field Olympus BX63 light microscope (OLYMPUSBX63, Olympus Corporation, Tokyo, Japan) at 10× and 40× magnification to observe microscopic changes in the diameter of the seminiferous tubule (ST) and lumen (LD), epithelium height (EH) and number of seminiferous tubules per field (no. ST/field). The height of the seminiferous epithelium was determined by the mean of two diametrically opposed measures EH1 and EH2. While the luminal tubular diameter was determined by the difference between the total diameter (TD) and the sum of both measures of the epithelium height (EH1 + EH2). Ten histological fields were counted on each testicular region from each group using the Image J public domain software [31].

### 2.8. Spermatogenesis Analysis

The maturation of germ cells in the seminiferous epithelium was categorized according to the Johnsen score. It applies the numbers from 1 to 10 to a cross section of each tubule according to the following criteria: 10 = complete spermatogenesis and perfect tubules; 9 = many spermatozoa present with disorganized spermatogenesis; 8 = only a few spermatozoa present; 7 = no spermatozoa, but many spermatids present; 6 = only a few spermatids present; 5 = no spermatozoa or spermatids, but many spermatocytes present; 4 = only a few spermatocytes present; 3 = only spermatogonia present; 2 = no germ cells present, but Sertoli cells present and 1 = no germ cells and no Sertoli cells present. The mean Johnsen score was calculated by randomly selecting 10 seminiferous tubules per Yangzhou goose gander.

### 2.9. Statistical Analysis

Computations were carried out with SPSS (Version 20.0) and Graph Pad Prism (Version 5.0). All values were expressed as the mean ± standard error of the mean (SEM). The differences across groups were calculated with a one-way analysis of variance (ANOVA), followed by a Turkey’s post hoc test and two-way ANOVA by considering Bonferroni post-tests to compare the means of the replicates, where *p*-values of <0.05 were considered significant.

## 3. Results

### 3.1. Steps of Spermiogenesis in Seminiferous Epithelium 

The steps of spermiogenesis and cellular associations in the seminiferous epithelium of Yangzhou ganders are shown in Figure 1. Ten stages of spermiogenesis were identified on the basis of nuclear morphological and acrosomal differentiation changes. Virtually in all stages (I-IX) we could observe the morphological development of spermatogonia (Ad, Ap1, Ap2 and B), spermatocytes (L, Z, P and Dp) and spermatids development (1–10). Similarly in all stages (I-IX) of the seminiferous epithelium variations, virtually all types of germ cells are present in every stage with a few exceptions. The steps of spermatogenesis in Yangzhou ganders is depicted in the cycle map in which we proceeded horizontally from left to right on a particular row and then continued from the left to right in the upper row again until observing the stage of sperm release in the lumen. 

Step 1: a new generation of spermatids that have just emerged from the second meiotic division touching the luminal and subluminal border of the epithelium. These spermatids had an oval shaped nucleus containing a number of centrally located stained chromatin bodies. The faintly stained area represents the Golgi complex in the cytoplasm. A few proacrosomal granules can be seen in the Golgi complex during this stage.

Step 2: the nuclear chromatin begins to decondense and appear uniformly distributed in the nucleoplasm. A large centrally located mass of acrosomal granules was observed connecting to the peripheral junctions by thin filamentous chromatin material. Small stained proacrosomal granules were located in the cytoplasm between the plasma and nuclear membranes.

Step 3: stage 3 spermatids had spherical nuclei, but chromatin decondensation had advanced in this stage and fewer small clumps of the chromatin were present. Acrosomic granules were attached to the nuclear membrane in this stage. The cell became oval and the nucleus seemed inconspicuous in shape.

Step 4: in this stage, the chromatin of the spherical nucleus continues to decondense along the nuclear membrane. The central chromatin mass started disappearing and the nucleus became eccentric within the cytoplasm spermatogonia, Ap2 (pale type spermatogonia), B type B spermatogonia, L (leptotene primary spermatocytes), P (pachytene primary spermatocytes), Z (zygotene primary spermatocytes), Dp (diplotene primary spermatocytes), Pe (peritubular myoid cell) and S (Sertoli cells).

Step 5: now the pear shaped nucleus contained the granular nucleoplasm. The cell also became elongated in shape. 

Step 6: spermatids at this stage had a further elongated cytoplasm and the cell was in a wavy shape. The nucleus seemed to be dividing into two lobes. Light microscopy observations indicated that the spermatid nucleus during elongation is coiled within the cytoplasm.

Step 7: the cell at this stage had divided into two equal halves. At the apex, the head of the spermatid became tapered. The nuclear chromatin became deeply stained and condenses to become granulofilamentous. 

Step 8: the deeply stained spermatid further elongated in shape. The nucleus elongated and became inconspicuous accompanied by a decrease in the dense chromatin material. The nucleoplasm seemed to be vanished and the spermatid head was observed pointing out.

Step 9: the nucleus was absent at this stage and the maximum length of the spermatid was seen at this stage. There was a marked reduction in the cytoplasm of the cell. 

Step 10: the spermatid at this stage is cylindrical in shape and maintains a curvature accompanied with a cytoplasm that completely disappeared. This was the final stage of sperm before releasing in lumen.

### 3.2. Cellular Associations in the Seminiferous Epithelium of Yangzhou Goose Ganders

Nine cellular associations were found in the seminiferous epithelium of Yangzhou ganders shown in Figure 2. Each cellular association occupied a specific small area. Different tubular cross sections contained various cellular associations, e.g., some tubular cross sections contained all nine cellular associations. Morphological changes in earlier generations of spermatids were used in illustrating cellular associations. The presence of spermatogonia (Ad, Ap1, Ap2 and B) as well as leptotene, pachytene and zygotene spermatocytes were observed frequently in the seminiferous epithelium. Morphological changes in developing spermatids facilitate the identification and classification of various cellular associations. Abbreviations of germ cells are as follows: Ad (dark type spermatogonia), Ap1 (pale type spermatogonia), Ap2 (pale type spermatogonia), B type B spermatogonia, L (leptotene primary spermatocytes), P (pachytene primary spermatocytes), Z (zygotene primary spermatocytes, Dp (diplotene primary spermatocytes), Pe (peritubular myoid cell) and S (Sertoli cell). Scale bar = 20 µm.

### 3.3. Diameter of ST, LD, EH and No. of ST/Field

Histological diagrams in Figure 3A–D shows the method of measuring the diameter of ST, EH and LD. Figure 3E–G shows that there was a positive correlation between the diameter of the seminiferous tubules, epithelial height and lumen diameter. They followed the same ascending and descending pattern throughout the experimental period. While the number of seminiferous tubules per field in Figure 3H exhibited a descending pattern throughout the experimental period. There was a significant decrease in the no. of ST/field on day 200 as compared to day 181. On day 227 it continued to decline and reached 20 tubules per field. Graphical data show that the number of ST per field was negatively correlated to testis weight, ST diameter, EH and LD. 

### 3.4. Johnsen Score

The Johnsen score was analyzed as depicted by the criteria mentioned above. In Figure 4A–D, on day 181, all the birds did not seem to be fully matured, so the number of germ cells in the seminiferous epithelium was low, resulting in a low Johnsen score. On day 200, there was a slight increase in the Johnsen score. While on day 227, when birds were fully matured, the Johnsen score significantly increased between days 200 and 227. In the histological picture, in Figure 4I–L on day 227, the seminiferous epithelium seemed fully matured and got the highest Johnsen score as compared to days 200 and 181. 

## 4. Discussion

The present study is the first to our knowledge that elaborates the steps of spermiogenesis and the classification of the seminiferous epithelium cycle, which measures histological parameters of seminiferous tubules development in matured and active testis in Yangzhou goose ganders. Normal development of germ cells is inevitable for spermatogenesis to attain high fertility in ganders, which can be seen by observing their morphological development. Thus we can speculate that germ cells (spermatogonia, spermatocytes and spermatids) act as one of the important factors affecting fertility and ultimately producing high quality and a high quantity of semen. Ten steps of spermiogenesis and nine stages of seminiferous epithelium were observed in Yangzhou ganders. Periodic alterations in Yangzhou ganders’ seminiferous epithelium were similar to other birds and mammals. Four types of spermatogonia, dark type A (Ad), two pale types Ap1, Ap2 and B spermatogonia were identified. 

In histological sections of the Yangzhou ganders’ testis, the first five stages of spermatids development was facilitated by the acrosomal position while the later five steps were identified by nuclear morphological changes. Similar observations were seen in the quail [32]. In this study, morphological changes of germ cells development were recognized by using hematoxylin and eosin stained sections. In the domestic fowl, similar observations were carried out by [33]. The cap phase that characterizes mammalian spermiogenesis [34] during spermatids development was absent in all stages of the cycle in Yangzhou ganders. These reports are in accordance with the guinea fowl [35], drakes [36] and in the Japanese quail [37]. 

Nuclear morphological changes in spermatids started from step 5 at which the nucleus elongated and began to show a spiral shape in the later stages. Several stages of seminiferous epithelium were present in the single seminiferous tubule cross section. These observations were similar to the duck [36] and Japanese quail [37].

In this study Ad spermatogonia were undifferentiated stem cells that were present in all stages of the seminiferous epithelium and were mostly found near the basement membrane. A similar observation was found [16] in Japanese quail. It shows that similar types of spermatogonia are present in the goose and quail. All four types of spermatogonia were present in all stages of the seminiferous epithelium and were therefore not distinguished in the classification. Leptotene, pachytene, zygotene and diplotene primary spermatocytes were frequently observed during spermiogenesis in Yangzhou ganders. One of the most important findings was the presence of leptotene primary spermatocytes in all stages of spermiogenesis and their long life span. These results are in accordance with previous reports in the duck [36], guinea fowl [35], Japanese quail [16] and greater Rhea [38] where leptotene, pachytene and zygotene primary spermatocytes were observed in the seminiferous epithelium of the relevant birds. Secondary spermatocytes are rarely observed in the seminiferous epithelium because their life span is very short [39]. In most of the mammals, each cross section of the seminiferous tubule consists of a single stage of the cycle of the seminiferous epithelium. However, in birds, as in primate cross sections of the seminiferous tubule, several different (heterogeneous) cellular associations or stages of the seminiferous epithelium are displayed [40].

Aging is not only the factor for testis development in birds, several factors are also involved in it. One of the major factors is the Sertoli cell that is responsible for testicular development. Sertoli cells acts as a central regulator of testis development [41,42]. In the present histological study, measuring the number of Sertoli cells in the seminiferous epithelium was beyond the scope of the study.

Seminiferous epithelium with an increasing age corresponds to the reduction in the number of Sertoli cells and germs cell composing the seminiferous epithelium [43]. As the bird reached maturity and with the onset of the breeding season, various histological parameters of the seminiferous epithelium, i.e., the diameter of the seminiferous tubules (µm), epithelial height (µm) and lumen diameter (µm), increased. An increase in the germ cells population in the seminiferous epithelium is directly proportional to the ST diameter, lumen diameter and epithelial height. 

In the present study, the diameter of ST, EH and lumen diameter were positively correlated. With the progression of spermatogenesis, the testes weight accelerated, the diameter of the seminiferous tubules increased accompanied with higher EH and LD on days 200. However there was a slight decrease in the ST diameter, EH and LD on days 227. These findings are in accordance with Jamieson et al. [39] that its generally agreed that the seminiferous tubular diameter, epithelial height, testicular weight and spermatogenesis are positively correlated. The Johnsen score was observed higher in seminiferous tubules having optimum numbers of germs cells and was least in ST having fragmentary lumen. A declining pattern in the number of ST per field was observed with an advancement of spermatogenic activity. Initially on day 181, the no. of ST/field was 34. Then it showed a descending pattern and lowered to 25. The no. of ST/field was significantly higher on day 181 as compared to day 200. On day 227, it further declined. ST diameter enhanced with increase in testis size, so seminiferous tubules covered more area under 10× magnification on day 200 and 227 as compared to day 181. It can be speculated that the number of ST per field is inversely proportional to the testicular weight, diameter of ST, EH and LD.

## 5. Conclusions

In conclusion, ten steps of spermiogenesis and nine stages of seminiferous epithelium were identified in Yangzhou goose ganders. Four types of spermatogonia Ad, Ap1, Ap2 and B were recognized. Taking in consideration the steps of spermiogenesis, stages of seminiferous epithelium, testicular weight, diameter of ST, EH, LD, Johnsen score and the number of ST per field, Yangzhou ganders seemed in the culmination phase. This information could be used to determine the age at which the ganders become morphologically and functionally prepared for reproductive activity. From our previous and present study, it is clear that body maturation in Yangzhou ganders is different from testis maturation. The body matured at 160–170 days of age while the testis underwent complete sexual maturation at 227 days of age. Histological evaluation of the testis proved that the testis underwent sexual maturation in the summer months in Yangzhou ganders.

## Figures and Tables

**Figure 1 animals-10-00570-f001:**
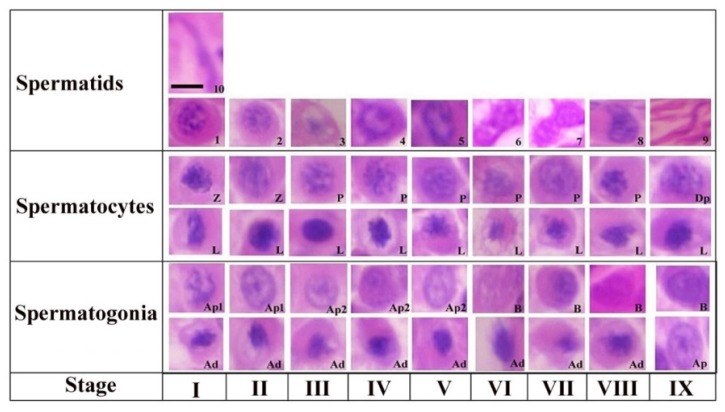
Stages of the seminiferous epithelial cycle in the Yangzhou goose ganders. Roman numerals indicate the stages of the cycle, whereas Arabic numerals indicate the steps of spermatids development (1–10). Ad (dark type spermatogonia), Ap1 (pale type spermatogonia), Ap2 (pale type spermatogonia), B type B spermatogonia, L (leptotene primary spermatocytes), P (pachytene primary spermatocytes) and D (diplotene primary spermatocytes). Scale bar = 1 µm.

**Figure 2 animals-10-00570-f002:**
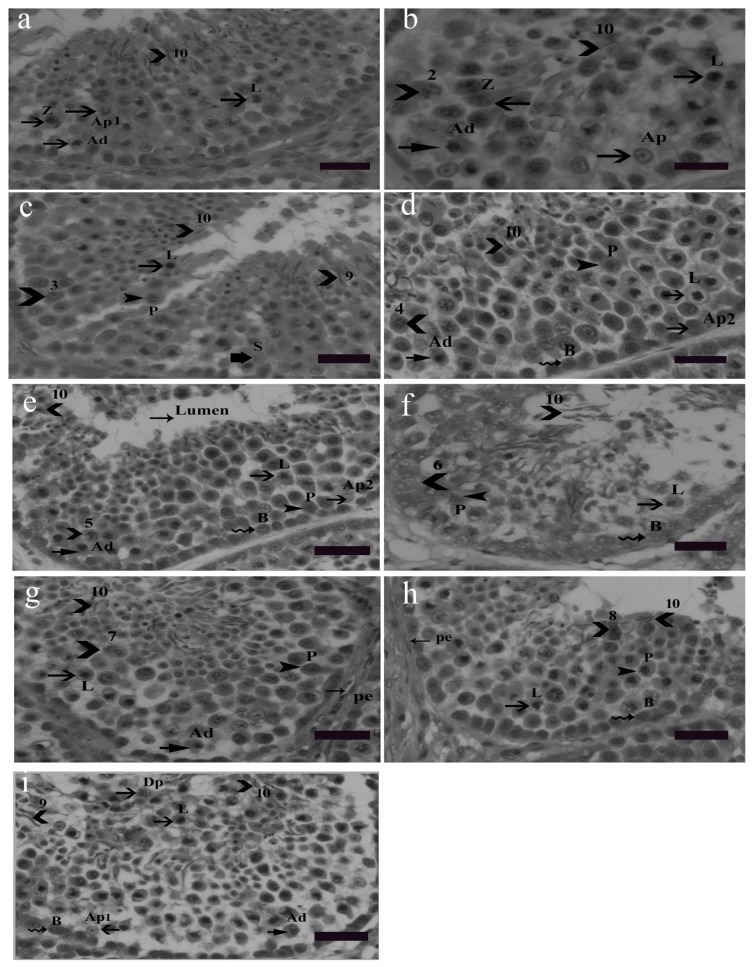
Stages 1–9 of the seminiferous epithelium cycle on the tubular morphology system. Abbreviations of germ cells are as follows: Ad (dark type spermatogonia), Ap1 (pale type spermatogonia), Ap2 (pale type spermatogonia), B type B spermatogonia, L (leptotene primary spermatocytes), P (pachytene primary spermatocytes), Z (zygotene primary spermatocytes, Dp (diplotene primary spermatocytes), Pe (peritubular myoid cell) and S (Sertoli cell). Scale bar = 20 µm. Stage 1: (**a**) this stage was identified by the presence of the earliest generation of spermatids at step 1 of spermiogenesis. Step 10 spermatids were clustered in groups embedded in the Sertoli cell cytoplasm. Primary spermatocytes were observed in the early leptotene and zygotene stages and they were scattered between spermatogonia and spermatids. During the development of spermatocytes, the nuclei of primary spermatocytes were morphologically larger and they contained dark stained chromatin strands twisted together. Two types of spermatogonia, the pale type Ap1 and dark type Ad, were located on the basement membrane. Type Ap1 spermatogonia were identified by the presence of chromatin particles having some larger and smaller clumps. Type B spermatogonia were identified by the presence of clumps of chromatin and hugging the basement membrane. Staining density of type B spermatogonia was intermediate between the dark Ad and pale type Ap1 spermatogonia. Stage 2: (**b**) as described above, stained granules were seen in the cytoplasm at stage 2. All other germ cells were the same as in stage 1 (**a**). Stage 3: (**c**) spermatids at this stage had become slightly oval in shape, lost their cytoplasm and became shortened. Mature spermatids were seen at this stage ready for spermiation. In the second row, leptotene spermatocytes were present whereas in the third and fourth row zygotene spermatocytes were observed. Spermatogonia were present near the basal membrane. Stage 4: (**d**) this stage was identified by the presence of step 4 spermatids. In the second row, leptotene primary spermatocytes were present. Step 10 spermatids were observed moving towards the luminal border. Between the third and fourth germ cell layers, pachytene spermatocytes were observed. Spermatogonia Ad, Ap2 and B lined the basal membrane. Stage 5: (**e**) the presence of Stage 5 spermatids confirmed this stage in which spermatids had a bigger acrosome, thickened nuclear membrane depicting a larger area of attachment than previous spermatids generations. The other germ cell types remained the same. Stage 6: (**f**) this stage was characterized by the existence of step 6 spermatids. Stage 6 spermatids exhibited specific nuclear morphological changes in which the cell nucleus almost vanished and the cell became elongated in shape. The spermatid head was observed protruding out oriented towards the basement membrane. Spermatogonia B lined near the basement membrane. Leptotene primary spermatocytes and pachytene primary spermatocytes were observed concomitantly in succeeding seminiferous epithelial layers. Stage 7: (**g**) in this stage the spermatids were observed in the cylindrical and more elongated shape having a spiral chromatin material wrapped along the stage 7 spermatid like helix. Stage 7 spermatids are slimmer than the previous stage spermatids. Due to acrosome elongation, cells seemed more wide morphologically. Pachytene, leptotene primary spermatocytes and spermatogonia Ad were observed as before in the subsequent layers. Stage 8: (**h**) the presence of stage 8 spermatids confirmed stage 8 of the seminiferous epithelium. The diameter of the cell as well as the nucleus further reduces. The nuclear chromatin condenses into coarse, round or oval shaped granules. Spermatids at this stage were observed in moving towards the lumen of the seminiferous epithelium. Other germs cells, i.e., primary spermatocytes in pachytene and leptotene stages were observed. Spermatogonia B lined the basement membrane. Stage 9: (**i**) at this stage spermatids were quite matured and hanging out near the lumen of the seminiferous epithelium. This stage was identified by the presence of stage 9 and 10 spermatids. These spermatids were observed in clusters and were less spiral in shape. First meiotic division gives rise to two haploid spermatocytes from a single diploid spermatocyte. Diplotene primary spermatocytes emerged as a result of the first meiotic cell division in which degradation of the synaptonemal complex can be observed. Leptotene primary spermatocytes form the next layer. Dark spermatogonia Ap1, pale spermatogonia Ad and B spermatogonia lined the basement membrane.

**Figure 3 animals-10-00570-f003:**
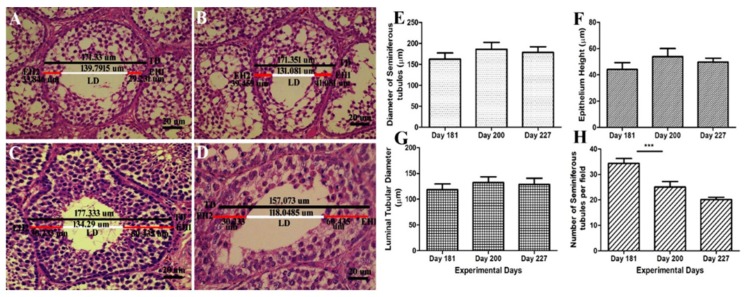
(**A**–**D**) Determination of the histometric measures of the seminiferous tubules from Yangzhou ganders testicles. TD—total tubular diameter; LD—luminal tubular diameter determined by the difference between the total diameter (TD) and the sum of both measures of epithelium height (EH1 + EH2); EH—height of the seminiferous epithelium determined by mean of two diametrically opposed measures (EH1 and EH2) and the scale bar represents 20 µm at 40× magnification. (**E**) Diameter of the seminiferous tubules (µm). (**F**) Epithelium height (µm). (**G**) Luminal tubular diameter (µm). (**H**) Number of seminiferous tubules per field (under 20× magnification). Data are shown as the mean values ± standard error of the mean, *** indicates difference *p* < 0.001, respectively between the groups.

**Figure 4 animals-10-00570-f004:**
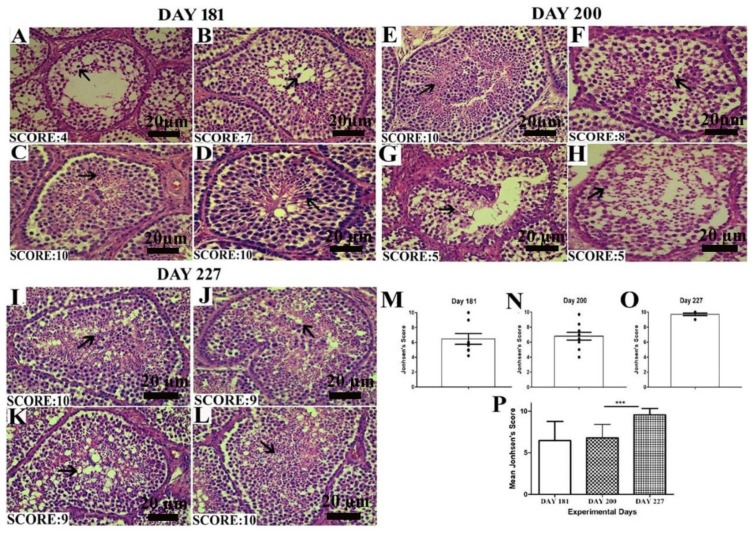
Images of histological sections on days 181 (**A**–**D**), 200 (**E**–**H**) and 227 (**I**–**L**) showing seminiferous tubules at different stages of spermatogenesis by the Johnsen score. Scale bar 20 µm at 40× magnification. (**M**–**O**) Graphical representation of the Johnsen score on days 181, 200 and 227 at a 95% confidence interval. (**P**) The mean Johnsen score. Data are shown as the mean values ± standard error of the mean, *** indicates difference *p* < 0.001, respectively between the groups. Scale bar = 20 µm.

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
