# Peer review of "Spermiogenesis, Stages of Seminiferous Epithelium and Variations in Seminiferous Tubules during Active States of Spermatogenesis in Yangzhou Goose Ganders"

_animals, 2020, doi:10.3390/ani10040570_

Round 1

Reviewer 1 Report

The new version of the manuscript has included the recommendations of the reviewers. However, I consider that the authors must make slight modifications on it before publication.

Simple Summary:

A general conclusion must be included from the results obtained in this study, in order to give  a recommendation to the livestock industry, e.g., when it is recommended to start the reproductive life in male goose (at 181 d, 200 d or 227d).

Abstract:

Pag       Line

1          35-36               Please, delete the sentence "Yangzhou goose has been developed by 8 generations of conventional breeding and 4 generations of pedigrees in 16 years”, because I do not understand the selection procedure and it is irrelevant for the results of this study.

1          37                     Please, change “Its popular in Chinese goose industry due to its high productivity and ……” to “Yangzhou goose is a synthetic breed developed by using local goose germplasm resources of China. Its popular in Chinese goose industry due to…”

1          48-52               Please, delete the sentence "Yangzhou goose is known as China's first new goose species having high reproductive efficiency. Yangzhou ganders reach sexual maturity between 160-170 days of age”, because it is not a result from your study.

Abstract must finish with a conclusion. For example, “….. Ours results suggest that complete sexual maturation is rearch at 227 days of age......”

Introduction:

The introduction is too long and too general in some parts.

Pag       Line

1          56-60               Please, delate “All goose species belong to the order of Anseriformes and the family of Anatidae[1]. Over the years, goose domestication, cross breeding, expanding demands for meat and eggs and modernization of civilizations turned it into profitable industry. Introduction of breeding programs added further momentum in waterfowl production. In the world goose production increased ten-fold between the 1960s and 2010, while in the same period a 17-fold increase occurred in Asia[2].”

2          74                    Please, delete “(2 years or older)”

3          112-128                       Please, replace  “Yangzhou goose(Anser domesticus) is synthetic breed developed by using local goose germplasm resources of China. It has been bred by 8 generations of conventional breeding and 4 generations of pedigrees in 16 years[31,32]. Professor Zhao Wanli a well-known expert in poultry genetics and breeding in China, also known as Father of Yangzhou geese[33] began to use Longchang goose that exhibit high growth rate and egg production[31,32]. The best combinations were selected for reverse cross and backcross, and then best combinations were chosen for generation breeding. The goslings for meat have robust growth rate, can feed on roughage, have strong adaptability, delicious meat, high muscle protein content, fine muscle fibers. It is rich in fat, low in water content, high processing yield, and good palatability. Under grazing and supplementary feeding system, the average live weight of 70-day-old goose is 3.49 kg and under control house-feeding system the average live-weight of the 70-day-old is 4.02 kg. Due to these economical and high yielding production qualities, more than 20 provinces such as Yan, Yancheng, Hongze, Wujin, Zhenjiang, Zhangjiagang, Dongtai, Xinghua, and Shandong, Henan, Shanghai, Heilongjiang, and Anhui have promoted the breeding of more than 100 million Yangzhou geese, which has led farmers to increase their income by nearly 800 million Yuan(114,560,000.00 USD). (Anser domesticus). The average egg production during breeding period is 77.6, natural mating and fertility rate reached 87.6-91.2%  [31-33] which is higher than other different goose breeds (53.8-84.72%) [34-37]. In past, all traits related to reproduction and fertility were addressed, except histological evaluation of germ cells development during spermatogenesis in Yangzhou ganders.  No systemic study is available regarding this unavoidable aspect. Therefore, keeping view all above fundamental aspects, the purpose of this study was to elaborate various stages of spermatogenesis showing germ cells development and cellular associations  in seminiferous epithelium of Yangzhou goose ganders. By depicting various stages and observing morphological changes in germ cells development, we can evaluate (I) efficiency of spermatogenesis (II) reproductive stage i.e. mature or immature testis and (III) histopathology of testis tissue, which may guide avian histopathologists, researchers and masses affiliated to goose industry to improve overall reproductive efficiency and fertility of ganders in specific”

To

“Yangzhou goose (Anser domesticus) is synthetic breed developed by using local goose germplasm resources of China [31,32]. This breed is widely used in China due to its high production traits for fertility, hatchability, egg production and meat production (references). This breed reaches the maturity weight at 160-170 days of age but the age at sexual maturity is unknown. The histological evaluation of germ cells development during spermatogenesis in Yangzhou ganders could you help us increase our knowledge in this field. Therefore, the aim of this study was examined the evolution in spermatogenesis in seminiferous epithelium of Yangzhou goose ganders at 180 d, 200 d and 227 d of age. By depicting various stages and observing morphological changes in germ cells development, we can evaluate (I) efficiency of spermatogenesis (II) reproductive stage i.e. mature or immature testis and (III) histopathology of testis tissue, which may guide avian histopathologists, researchers and masses affiliated to goose industry to improve overall reproductive efficiency and fertility of ganders in specific.  . ”

3          130      Please, delete “regarding this unavoidable aspect”

3          131      Please, change “purpose” to “objective”

3          131       Please change “elaborate” to “analyse”

Materials and Methods:

Ok

Results:

Pag      Line

5          121      Please, change “Spermiogenesis” to  “spermiogenesis”

Discussion:

Ok

Conclusions:

They must be obtained from your results. For that, please delete “ Yangzhou geese breed developed after 16 years of extensive breeding programs and is successfully contributing in Chinese economy due to its high production traits for fertility, hatchability, egg production and meat production.  For optimize goose production, improvements in ganders reproductive efficiency is equally important to nutritional requirements, disease control and housing” because this affirmation is not obtain from your study.

Author Response

Simple summary:

                             A general conclusion must be included from the results obtained in this study, in order to give  a recommendation to the livestock industry, e.g., when it is recommended to start the reproductive life in male goose (at 181 d, 200 d or 227d).

Reply: Conclusion, " In Yangzhou goose ganders, recommended age of sexual maturity is 227 days of age (lines 29-30 in revised manuscript)

Page       Line

1             35-36

Please, delete the sentence "Yangzhou goose has been developed by 8 generations of conventional breeding and 4 generations of pedigrees in 16 years”, because I do not understand the selection procedure and it is irrelevant for the results of this study

Reply: Deleted( lines 37-38 in revised manuscript)

1            37                     Please, change “Its popular in Chinese goose industry due to its high productivity and ……” to “Yangzhou goose is a synthetic breed developed by using local goose germplasm resources of China. Its popular in Chinese goose industry due to…”

Reply: Corrected (Yangzhou goose is synthetic breed developed by using local goose germplasm resources of China. It's popular in Chinese goose industry due to high productivity and adaptability lines 39,40,41 in revised manuscript)

1          48-52               Please, delete the sentence "Yangzhou goose is known as China's first new goose species having high reproductive efficiency. Yangzhou ganders reach sexual maturity between 160-170 days of age”, because it is not a result from your study

Reply: Deleted (same lines 52, 53, 54 in revised manuscript)

Abstract

Abstract must finish with a conclusion. For example, “….. Ours results suggest that complete sexual maturation is reach at 227 days of age....

Reply: Abstract finished with conclusion, " Our results suggests that Yangzhou ganders reach complete sexual maturity at 227 days of age. ( lines 56, 57 in revised manuscript)

Introduction

Page        line

1            56-60          Please, delete “All goose species belong to the order of Anseriformes and the family of Anatidae[1]. Over the years, goose domestication, cross breeding, expanding demands for meat and eggs and modernization of civilizations turned it into profitable industry. Introduction of breeding programs added further momentum in waterfowl production. In the world goose production increased ten-fold between the 1960s and 2010, while in the same period a 17-fold increase occurred in Asia[2].”

Reply: Deleted( lines 60-65 in revised manuscript)

2             74               Please, delete “(2 years or older)

Reply:   78             ( in revised manuscript, deleted)

3           112-128       (same lines 117-142 in revised manuscript)

Please, replace  “Yangzhou goose(Anser domesticus) is synthetic breed developed by using local goose germplasm resources of China. It has been bred by 8 generations of conventional breeding and 4 generations of pedigrees in 16 years[31,32]. Professor Zhao Wanli a well-known expert in poultry genetics and breeding in China, also known as Father of Yangzhou geese[33] began to use Longchang goose that exhibit high growth rate and egg production[31,32]. The best combinations were selected for reverse cross and backcross, and then best combinations were chosen for generation breeding. The goslings for meat have robust growth rate, can feed on roughage, have strong adaptability, delicious meat, high muscle protein content, fine muscle fibers. It is rich in fat, low in water content, high processing yield, and good palatability. Under grazing and supplementary feeding system, the average live weight of 70-day-old goose is 3.49 kg and under control house-feeding system the average live-weight of the 70-day-old is 4.02 kg. Due to these economical and high yielding production qualities, more than 20 provinces such as Yan, Yancheng, Hongze, Wujin, Zhenjiang, Zhangjiagang, Dongtai, Xinghua, and Shandong, Henan, Shanghai, Heilongjiang, and Anhui have promoted the breeding of more than 100 million Yangzhou geese, which has led farmers to increase their income by nearly 800 million Yuan(114,560,000.00 USD). (Anser domesticus). The average egg production during breeding period is 77.6, natural mating and fertility rate reached 87.6-91.2%  [31-33] which is higher than other different goose breeds (53.8-84.72%) [34-37]. In past, all traits related to reproduction and fertility were addressed, except histological evaluation of germ cells development during spermatogenesis in Yangzhou ganders.  No systemic study is available regarding this unavoidable aspect. Therefore, keeping view all above fundamental aspects, the purpose of this study was to elaborate various stages of spermatogenesis showing germ cells development and cellular associations  in seminiferous epithelium of Yangzhou goose ganders. By depicting various stages and observing morphological changes in germ cells development, we can evaluate (I) efficiency of spermatogenesis (II) reproductive stage i.e. mature or immature testis and (III) histopathology of testis tissue, which may guide avian histopathologists, researchers and masses affiliated to goose industry to improve overall reproductive efficiency and fertility of ganders in specific”

Reply: Deleted

3-4                   144-156         lines in revised manuscript added

3                       130                   Please, delete “regarding this unavoidable aspect”

Reply:                                       Deleted

3                       131                 Please, change “purpose” to “objective

Reply: 4            150               "Purpose" changed to "objective"( in revised manuscript)

3                         131                 Please change “elaborate” to “analyze”

Reply:4            150                 "elaborate" changed to "analyze" (in revised manuscript)

Materials and Methods

                                 ok

Results

Page                 Line

5                       121                   Please, change “Spermiogenesis” to  “spermiogenesis”

Reply:5             231                   "Spermiogenesis" changed to "spermiogenesis"

Discussion

                               ok

Conclusions

They must be obtained from your results. For that, please delete “ Yangzhou geese breed developed after 16 years of extensive breeding programs and is successfully contributing in Chinese economy due to its high production traits for fertility, hatchability, egg production and meat production.  For optimize goose production, improvements in ganders reproductive efficiency is equally important to nutritional requirements, disease control and housing” because this affirmation is not obtain from your study.

Reply:14           450-454                   Deleted

Reviewer 2 Report

Comments to authors

The significance of this study (title: Spermiogenesis, stages of seminiferous epithelium and variations in seminiferous tubules during active states of spermatogenesis in Yangzhou goose ganders) is to divide the seminiferous epithelium cycle stages in Yangzhou ganders. But there are still some problems that need to be solved.

In line46: embedded in epoxy resins; In 198: molten paraffin wax. Checking consistency.

Too much business information for goose. In introduction, some tedious information is unnecessary and meaningless.

Line269: stage1 should be changed to step1.

Results 3.1 did not fully show the results. Figure 1 needs to be introduced one by one in results section. Otherwise, readers cannot understand what to express in figure 1. Figure 1 seems to just piece together some screenshots. What is the meaning and role of Figure 1?

The division of the seminiferous epithelium cycle is based on which previous study. In results 3.2, the author only refers to listing some characteristics of each step, but there is no evidence and detail pictures to prove it. The author must supplement the figures for each stages of seminiferous epithelium cycle. Pictures of seminiferous epithelium cycle should be displayed in accordance with previous research (DOI: 10.1095/biolreprod.106.053835; DOI: 10.1095/biolreprod.102.010652; DOI: 10.1016/j.theriogenology.2004.09.014; DOI: 10.1095/biolreprod67.1.247).

The quality of writing needs to be improved. Authors should check the manuscript carefully. The sentence of results should indicate which picture the description matches (figure 3A or 3B?). And, where are parts of (1), (2), (3), (4) in Figure 2?

Please check and double-check when preparing your manuscript before submitting.

Author Response

  1. In line46: embedded in epoxy resins; In 198: molten paraffin wax. Checking consistency

Reply: line 44       " embedded in epoxy resins" changed to " embedded in molten paraffin wax"

                               in revised manuscript.

Introduction

  1. 2. Too much business information for goose. In introduction, some tedious information is unnecessary and meaningless.

Reply: Page 2 lines 60-65       deleted (in revised manuscript)

           Page 3 lines 117-143   deleted (in revised manuscript)

           Page 3-4, lines 144-156 added (in revised manuscript)

Results

  1. Line269: stage1 should be changed to step1.

Reply: Page 6, line 247: " Stage 1 changed to Step 1 ( in revised manuscript)

  1. Results 3.1 did not fully show the results. Figure 1 needs to be introduced one by one in results section. Otherwise, readers cannot understand what to express in figure 1. Figure 1 seems to just piece together some screenshots. What is the meaning and role of Figure 1?

Reply: Page 5-6, lines 234-237 added in revised manuscript "Vertically in all stages (I-IX) we can observe morphological development of spermatogonia (Ad, Ap1, Ap2, B), spermatocytes (L, Z, P, Dp) and spermatids(1-10). Similarly in all stages (I-IX) of seminiferous epithelium variations, vertically all types of germ cells are present in every stage with a few exceptions". It shows 9 stages of cellular associations of seminiferous epithelium of Yangzhou ganders in roman numerals (I-IX) while in Arabic numbers(1-10) it shows morphological maturation of spermatids. Similarly its shows spermatogonia and spermatocytes development in all nine stages (I-IX) of seminiferous epithelium variations during spermiogenesis. It can also be noted that for example vertically in stag I ( spermatogonia Ad,Ap1 spermatocytes L , Z and spermatids 1,10) are all present with exception of one or two cells. Similar for stage II, III up to IX. Role and purpose of figure 1 is to show morphological development of spermatogonia, spermatocytes and spermatids development

  1. 5. The division of the seminiferous epithelium cycle is based on which previous study. In results 3.2, the author only refers to listing some characteristics of each step, but there is no evidence and detail pictures to prove it. The author must supplement the figures for each stages of seminiferous epithelium cycle. Pictures of seminiferous epithelium cycle should be displayed in accordance with previous research (DOI: 10.1095/biolreprod.106.053835; DOI: 10.1095/biolreprod.102.010652; DOI: 10.1016/j.theriogenology.2004.09.014; DOI: 10.1095/biolreprod67.1.247).

Reply: division of the seminiferous epithelium cycle is based on https://doi.org/10.1139/cjas-2016-0068. and

doi: 10.1292/jvms.14-0411; J. Vet. Med. Sci. 77(7): 799–807, 2015

  1. 6. The quality of writing needs to be improved. Authors should check the manuscript carefully. The sentence of results should indicate which picture the description matches (figure 3A or 3B?). And, where are parts of (1), (2), (3), (4) in Figure 2?

Reply: Figure 2 is divided into 2a(1-4) and 2b(5-9) In figure 3, descriptions are added in revised manuscript.

Reviewer 3 Report

This manuscript clearly identified ten steps of spermiogenesis and nine stages of seminiferous epithelium of Yangzhou goose ganders by histological observation of germ cells development and cellular associations in seminiferous epithelium. The results of the study revealed that initial steps of spermiogenesis were depicted by changes in acrosomic granules, whereas further stages were identified by nuclear morphological changes. Furthermore, four types of spermatogonia Ad, Ap1, Ap2 and B were recognized during spermiogenesis.

The authors did a good and meaningful job. But there are several suggestions for minor improvement.

  1. The figure resolution in this manuscript need to be improved.
  2. In the introduction part, the author emphasized that the results of this study may guide researchers and masses affiliated to goose industry to improve overall reproductive efficiency and fertility of ganders in specific, but there is a lack of discussion in this regard in the discussion section.
  3. Please add scale bar in figure 1, 2 and 4.
  4. There are some spilling mistakes (e.g., testis and testes).

Author Response

  1. The figure resolution in this manuscript need to be improved

Reply: The slides of testis sections were already observed under highest magnifications of 10X and 40X under bright field Olympus BX63 light microscope (OLYMPUSBX63). Increasing figures resolution will break its pixels resulting in blur figures. Secondly observing testis slides under more high resolution light microscope is beyond scope of study.

  1. 2.     In the introduction part, the author emphasized that the results of this study may guide researchers and masses affiliated to goose industry to improve overall reproductive efficiency and fertility of ganders in specific, but there is a lack of discussion in this regard in the discussion section

Reply: In revised manuscript, DISCUSSION part, lines 389-393 are added explaining this aspect.

  1. Please add scale bar in figure 1, 2 and 4.

Scale bars added in figure 1, 2 and 4 in revised manuscript.

  1. 4. There are some spelling mistakes (e.g. testis and testes)

Reply: In whole revised manuscript, spelling mistakes including (testis and testes) are corrected.

Round 2

Reviewer 2 Report

Comments to authors

The title of Results 3.2 is missing. Please add.

Previous comments, we suggested that “The author must supplement the figures for each stages of seminiferous epithelium cycle. Pictures of seminiferous epithelium cycle should be displayed in accordance with previous research (DOI: 10.1095/biolreprod.106.053835; DOI: 10.1095/biolreprod.102.010652; DOI: 10.1016/j.theriogenology.2004.09.014; DOI: 10.1095/biolreprod67.1.247)”. These are classic research reports. BUT, the authors completely ignore this comment without any reply or modification.

Author Response

1. Title of Results 3.2 is missing. Please add
Reply. Title of results sequence 3.1, 3.2, 3.3, 3.4 corrected
2. Previous comments, we suggested that “The author must supplement the figures for each stages of seminiferous epithelium cycle. Pictures of seminiferous epithelium cycle should be displayed in accordance with previous research (DOI: 10.1095/biolreprod.106.053835; DOI: 10.1095/biolreprod.102.010652; DOI: 10.1016/j.theriogenology.2004.09.014; DOI: 10.1095/biolreprod67.1.247)”. These are classic research reports. BUT, the authors completely ignore this comment without any reply or modification.
Reply: The figures of seminiferous epithelium cycle are modified according to the references recommended by the reviewer, however, details of every two steps are given one after other because the third reviewer demanded Figure 2 to be in high resolution, if we plot Figure 2 exactly in same manner as in above references, Figure 2 would be small and of low picture resolution. Thats why every two stages of seminiferous epithelium are pasted and described step by step.

Reviewer 3 Report

The image resolution in this version has been greatly improved.

Author Response

Thanks for your suggestion.

This manuscript is a resubmission of an earlier submission. The following is a list of the peer review reports and author responses from that submission.

Round 1

Reviewer 1 Report

In this study, spermiogenesis and morphological changes in the seminiferous epithelium are evaluated in Yangzhou goose from 181 days until 220 days of age. There is a great workload and the manuscript has nice photos. However, the manuscript seems to be merely a descriptive work of the spermiogenesis in Yangzhou goose without explaining sufficiently the interest and/or impact on goose industry.

In this sense, for example, it would have been interesting to evaluate the fertilizing capacity of sperm in the different age groups (181 d, 200 d and 220 d). Besides, the body weight of animal may have importance in the development of the spermatozoa, then why this variable was not including as covariance in the study.

For all these reasons, I recommend that the authors rewrite the manuscript taking into account these remarks before again submitting it.

Reviewer 2 Report

Regarding the manuscript submitted by Akhtar et al. determined morphological changes of germ cells of Yangzhou goose ganders during spermiogenesis using morphological analysis (hematoxylin and eosin stained testis sections). They found that “ten steps of spermiogenesis, nine stages of seminiferous epithelium and four types of spermatogonia: Ad, Ap1, Ap2 and B.”

However, this manuscript is not suitable for publication in its current form. As presented, the background is vague, there are numerous English grammar and style errors, and the images of testis morphology are unclear. Suggested revisions listed below.

Major

Correct the English grammar and style errors. Specifically, the authors used “spermatogenesis development” in text, the authors should state “testis development” or “spermatogenesis.” I strongly recommend the authors submit their manuscript for editing by English and grammar specialists. Pages 2, Lines 63-70: The authors examined germ cell differentiation during spermiogenesis. However, they didn’t mention why spermiogenesis is important for Yangzhou goose ganders. Also, the experimental rational for using the Yangzhou goose gander was not provided.. Please state the reason in the Abstract and Introduction. Why did the authors select an age of 161 days? Why were samples collected on days 181, 200, 227? Please provide these explanations in the Materials and Methods. Images: Testicular morphology is the most important part of the manuscript. Despite this, all images are distorted. For example, the authors captured the images with a lower magnification in Figure 2. It is hard to see which germ cells the authors pointed out. If the authors provided the images of specific-germ cells with higher magnification, readers would be able to clearly and accurately see the author’s examples.

Minor

Please add country name of the source in Materials and Methods. Line 50: “and 10 days in bandicoot rats is…..” is incomplete sentence. Please correct it.